# Maternity care providers' perspectives on late-term gestation management (LATE-study): A cross-sectional survey

Anna E. Seijmonsbergen-Schermers[1,2,3,4]*, Fatima Hammiche[5,6], Hannah de Klerk[1,2,3,4], Judit K.J. Keulen[7], Caroline J. Bax[5,6], Lilian L. Peters[1,2,3,4], Corine J. Verhoeven[1,2,4,6,8,9]

**1** Department of Midwifery Science, Amsterdam UMC Location Vrije Universiteit Amsterdam, Amsterdam, the Netherlands, **2** Midwifery Academy Amsterdam Groningen, Inholland, Amsterdam, the Netherlands, **3** Amsterdam Public Health, Quality of Care, Amsterdam, the Netherlands, **4** Department of Primary and Long-Term Care, University of Groningen, University Medical Center Groningen, Groningen, the Netherlands, **5** Department of Obstetrics and Gynaecology, Amsterdam UMC Location Amsterdam University Medical Center, Amsterdam, the Netherlands, **6** Amsterdam Reproduction and Development, Amsterdam, the Netherlands, **7** Research Centre for Midwifery Science, Zuyd University of Applied Sciences, Maastricht, the Netherlands, **8** Department of Obstetrics and Gynaecology, Maxima Medical Centre, Veldhoven, the Netherlands, **9** Division of Midwifery, School of Health Sciences, University of Nottingham, Nottingham, United Kingdom

\* a.seijmonsbergen@amsterdamumc.nl

## Abstract

### Objective

To examine the experiences, perspectives, and impact on daily work of maternity care providers regarding the updated Dutch national guideline on the management of late-term pregnancies.

### Design and setting

This was a cross-sectional survey for maternity care providers in the Netherlands in 2024.

### Methods

An online questionnaire was developed, improved on the basis of four semi-structured interviews, piloted, and distributed to primary care and hospital-based midwives, residents, and obstetricians. A 5-point Likert Scale was used for most questions. Descriptive analyses and data visualisation were performed. Linear regressions were performed to assess the association between characteristics and agreement with and adherence to the recommendations. The main outcome measures were awareness of the guideline, agreement with and adherence to the recommendations, opinions and perspectives on the impact of the guideline, and experiences of participants regarding the guideline.

**Data availability statement:** All files are available from the DANS data Station Life Sciences (https://doi.org/10.17026/LS/KZMWQT).

**Funding:** This study was funded by the Koninklijke Nederlandse Organisatie van Verloskundigen, the Dutch Professional Organization of Midwives (project number 2012862). In addition, there was internal funding from Inholland University of applied sciences, Midwifery Academy Amsterdam Groningen (AVAG). The corresponding author had full access to all the data in the study and had final responsibility for the decision to submit for publication. The funders had no role in study design, data collection and analysis, decision to publish, or preparation of the manuscript.

**Competing interests:** Hannah de Klerk (midwife) and Caroline Bax (obstetrician) were members of the working group who developed the Dutch national guideline on management of pregnancies at 41 weeks' gestation onwards. This does not alter our adherence to PLOS ONE policies on sharing data and materials. All authors have declared that no other competing interests exist.

## Results

A total of 643 care providers were included, of whom 53% were primary care midwives, 29% obstetricians/residents and 19% hospital-based midwives. There was 72% agreement and 87% adherence to "offer induction of labour (IOL) at 41 weeks alongside expectant management". Although 92% agreed with "discuss the advantages and disadvantages", only 67% agreed with "specify perinatal outcomes" and 63% agreed with "specify outcomes for nulliparous versus multiparous women". Adherence to these and to recommendations regarding monitoring were lower (43–59%). There was uncertainty among 45% about the benefits of the guideline, with only 17% believing in maternal and 20% believing in neonatal benefits.

## Conclusions

Care providers widely agreed with and adhered to the recommendation to offer IOL at 41 weeks, but adherence to specific elements of shared decision-making was lower, and care providers expressed limited confidence on the benefits of the policy for mother and child.

## Introduction

In recent years, induction of labour (IOL) has been increasingly performed to reduce maternal and perinatal mortality and morbidity [1–5]. In addition to pregnancy complications such as hypertensive disorders, intrauterine growth restriction, and intrahepatic cholestasis [6], maternal and pregnancy characteristics are increasingly being considered as indications for IOL [6,7]. Gestational age is another important indication for IOL that applies to many women [6,8,9].

In 2020, a systematic review on IOL at 41 weeks versus expectant management (EM) up to 42 weeks in low-risk women was published [10]. The meta-analysis found a significantly lower incidence of the composite adverse perinatal outcome in nulliparous women in the IOL group compared to the EM group, with no significant differences in mode of birth [10].

The results of the studies included in this meta-analysis have led to changes in policy for late-term gestations in many countries [11–16]. In the Netherlands, IOL was previously recommended at 42 weeks. In 2021, a new guideline on management of late-term gestations was developed in collaboration with the professional associations of midwives, obstetricians, paediatricians, and client representatives [16]. This guideline was based on the results of the INDEX [17] and the SWEPIS trials [18]. The meta-analysis had not yet been published at the time the guideline was developed, but a meta-analysis was performed for the purpose of the guideline.

This guideline was aimed at Dutch maternity care providers and consists of four recommendations in two modules. Module 1 considers the option of IOL at 41 weeks versus EM (Box 1), while module 2 considers additional checks after 41 weeks if a woman chooses EM (Box 2).

The guideline working group on late-term management noted that the feasibility and acceptability of the recommendations were not yet known. They expressed concern that the new policy may raise issues related to hospital capacity and the ability to provide continuous support during labour [16]. This study aims to explore the experiences and perspectives of maternity care providers on the management of late-term gestations outlined in the guideline. Moreover, it is currently unclear to what extent the guideline is being adhered to. The guideline may also influence daily work of maternity care providers [19], who often face capacity challenges. Understanding adherence and the perspectives and experiences of care providers is crucial to refining the guideline and incorporating valuable feedback from care providers.

This aim of this study is therefore to examine the experiences, perspectives, and impact on daily work of maternal care providers regarding the updated Dutch national guideline on the management of late-term pregnancies and its recommendations.

## Materials and methods

This cross-sectional survey study is part of the LATE-study: LATE-term gestation management. This study examines both midwife-led and obstetrician-led care in the Netherlands. Box 3 describesthe organisation of maternity care in the Netherlands [20,21]. The target population consisted of primary care midwives, hospital-based midwives, residents, and obstetricians. Care providers who had not worked for more than one year were excluded from this study. Participants were excluded from analyses if less than 25% of the questionnaire was completed.

*The Dutch maternity care system is based on a division between primary care provided in the community and secondary and tertiary care in hospitals [20]. Women's care is based on the assessment of the individual risk of each woman. Women with a low-risk pregnancy are cared for by community midwives in midwife-led, primary care, and have the option of birthing at home, in a birth centre or to have a midwife-led hospital birth. Women at intermediate or high-risk are referred to obstetrician-led secondary or tertiary care where they are looked after by hospital-based midwives, nurses and obstetricians [21].*

The questionnaire was developed by the authors and a client representative. During the development phase, four qualitative semi-structured interviews were held with midwives and obstetricians to explore themes that were important to care providers regarding the guideline on management of late-term gestations and that should be incorporated into the questionnaire. In advance, written informed consent was obtained. These interviews were analysed using inductive thematic analysis. Different versions of the questionnaire were discussed by the working group and then piloted by a total of twelve care providers and researchers. During the pilot phase, some items were added to the questionnaire, and others were removed or clarified.

The final version of the questionnaire was incorporated into Castor EDC [22] and ran from November 30th, 2023 until June 4th, 2024. The online questionnaire was disseminated by the professional organizations of midwives and obstetricians through newsletters and social media. The authors of this paper also shared the questionnaire through social media and their contacts. In addition, the Federation of Medical Specialists in the Netherlands forwarded the questionnaire to the email addresses of all regional midwifery and obstetrics collaborations. Furthermore, at several congresses for maternity care providers, participants were invited to complete the questionnaire.

The questionnaire began with a request for written informed consent within the questionnaire. Participants were first provided with ethical information about the study procedure, followed by a question asking whether they agreed with the information and consented to participate.

The questionnaire comprised seven topics (see Supporting information 1):

1) General information about the study and the guideline.

2) Awareness of the existence of the guideline recommendations.

3) Opinion and adherence to the recommendations in the guideline.

4) Shared decision-making: adherence to the recommendations on shared decision-making in the guideline and content of the conversations.

5) Expected outcomes and impact of the recommendations on practice.

6) Characteristics of participants.

7) General attitude towards birth as natural or medical using the validated Birth Beliefs Scale (BBS) [23].

In this paper we focused on opinion and adherence to the recommendations and the expected outcomes and impact of the recommendations on practice. Awareness, characteristics, and attitude were considered as characteristics of the participants in this study. In another paper, we described the results of the questions on shared-decision making [24].

The Birth Beliefs Scale (BBS) is a validated Dutch measure for maternity care providers and consists of eleven items [23]. The BBS measures the concept of viewing birth as medical or natural and comprises two domains: a medical view, maintaining that childbirth is potentially pathological, and a natural view, maintaining that childbirth is a largely physiological event that occurs in most women's lives. The scale contains five statements concerning the medical view and six statements concerning the natural view. Participants were asked to rate their level of agreement on a 5-point Likert scale (from 'strongly disagree' to 'strongly agree' [23].

## Analyses

Characteristics of participants that applied to a specific profession or group were described as numbers and percentages for categorical variables and as means and standard deviations (SD) for continuous variables. Characteristics applying to all participants were described as numbers and percentages or means and SD for continuous variables or Likert scales, stratified by three professional groups: obstetricians and residents, hospital-based midwives, and primary care midwives. The aim of stratifying by professional group was to provide clear and meaningful results, as we expect the professional

groups not being comparable. Presenting results for the whole group would be heavily influenced by the professional group with the highest representation in the survey. Opinions of care providers on management of late-term gestations and the impact of the policy in the guideline, rated on a 5-point Likert scale, were shown using data visualisations (stacked bar graphs), visually displaying the percentages of the Likert scales rated by participants. For this visualization, the categories 'strongly disagree/never' and 'agree/rarely' were combined, as well as the categories 'agree/often' and 'strongly agree/always'. These results were then presented in line bars, showing the mean of the 5-point Likert scores for each of the three profession groups described above.

Linear regression analyses were then performed to assess the association between the characteristics and the agreement with and adherence to the recommendations of module 1. This resulted in a regression coefficient of agreement and adherence for each characteristic for the recommendations R1.1, R1.2a, R1.2b, R1.2c, and R1.2d. The assumptions of linearity and normality of the residuals of outcomes were checked beforehand. For the independent variable region, the weighted mean was taken as the reference category. Finally, the preferred policy option of care providers who disagreed with offering IOL at 41 weeks was presented in numbers and percentages, stratified by professional group.

Data analyses were conducted in STATA version 14.1 (StataCorp, Texas, USA). The Amsterdam University Medical Center confirmed ethical approval for this study (reference 2023.0339).

## Results

### Participant characteristics

A total of 497 care providers completed all items of the online questionnaire. An additional 146 care providers completed at least 25% of the questionnaire and were included in the analyses.

Of the 643 included participants, 53% worked as a primary care midwife, 22% as an obstetrician, 19% as a hospital-based midwife, and 7% as a resident (Tables 1 and 2). The majority of obstetricians (90%) worked in an obstetric unit, most of which were in large hospitals, including 23% in tertiary care centres. Most primary care midwives worked in a practice with 3–5 midwives (58%). Most midwives were female (99%), while 74% of the obstetricians were female. The mean age of the participants was 41 years (SD 11), with 16 years (SD 10) of working experience. Hospital-based midwives were more often trained abroad (37%), compared to primary care midwives (15%) and obstetricians (2%). Hospital-based midwives were typically paid employed (98%), compared to 55% of obstetricians and just 8% of primary care midwives.

Geographically, most participants were from the Southwest Netherlands, with a small representation from Limburg. All professions scored above a score of 3 on a 5-point Likert scale (representing a neutral score) for knowledge about the scientific foundation of the guideline (mean 3.8), positive attitude towards guidelines (3.7), and experience of capacity problems (3.6). On the Birth Belief Scale, obstetricians had similar scores for medical and natural views (3.3), whereas midwives scored higher on the natural view (3.7 for hospital-based midwives and 4.2 for primary care midwives).

Awareness of the guideline modules varied: module 1 was well-known to participants (92%), whereas module 2 was less well known (68%). A large proportion of participants who were familiar with the guideline did not read beyond the recommendations (34% for module 1 and 40% for module 2), with a small subgroup not reading the module 1 (5%) and module 2 (6%) at all.

### Agreement and adherence

Overall, the majority of participants agreed with the recommendations (Fig 1a), with obstetricians showing higher levels of agreement than midwives (Fig 1b). Of all participants, 72% agreed with the recommendations to offer IOL at 41 weeks alongside the option of EM (R1.1). Of those who disagreed, 84% were primary care midwives. Of all participants who disagreed, the majority (70%) felt that IOL should be delayed until 42 weeks (Table 3). Although 92% agreed with the recommendation to discuss the advantages and disadvantages (R1.2d), only 67% agreed with the recommendation to specify perinatal outcomes (R1.2b) and only 63% agreed with the recommendation to specify outcomes for nulliparous versus

**Table 1. Participant characteristics, applying to a specific profession or group.**

| Variables | n (%) or mean [SD] |
|---|---|
| Total n | **Total n = 643** |
| Profession | Total n = 643 |
| *Obstetrician* | 139 (22) |
| *Obstetric resident* | 43 (7) |
| *Hospital-based midwife* | 120 (19) |
| *Primary care midwife* | 341 (53) |
| Applying to obstetricians: | Total n = 123 |
| Main speciality | |
| *Perinatology* | 95 (77) |
| *Urogynaecology* | 5 (4) |
| *Reproductive medicine* | 4 (3) |
| *Oncology* | 2 (2) |
| *Benign gynaecology* | 6 (5) |
| *General obstetrician* | 11 (9) |
| Applying to obstetricians and obstetric residents: | Total n = 155 |
| Frequency work at an obstetric unit | |
| *Weekly* | 139 (90) |
| *Bi-weekly* | 13 (8) |
| *Monthly* | 2 (1) |
| *Less than monthly* | 1 (1) |
| *Never* | |
| Applying to obstetrician-led care: | Total n = 244 |
| Hospital type | |
| *Tertiary/academic center* | 57 (23) |
| *Secondary/peripheral center* | 187 (77) |
| Hospitals' number of births per year | |
| *<500* | 1 (0) |
| *500-1000* | 18 (7) |
| *1000-1500* | 38 (16) |
| *1500-2000* | 46 (19) |
| *>2000* | 133 (55) |
| *Unsure or missing* | 7 (3) |
| Applying to midwives: | Total n = 349 |
| Highest level of education | |
| *Bachelor degree* | 283 (81) |
| *Master degree* | 56 (16) |
| *PhD* | 10 (3) |
| Applying to primary care midwives: | Total n = 259 |
| Practice size | |
| *1-2 midwives* | 30 (12) |
| *3-5 midwives* | 151 (58) |
| *6-10 midwives* | 70 (27) |
| *11 or more midwives* | 8 (3) |
| Transfer to hospital of first choice is possible (on scale from 1 to 10), mean [SD] | 6.8 [2.1] |

Number of missings: n = 0 for profession, n = 16 for questions applying to obstetricians, n = 27 for questions applying to obstetricians and obstetric residents, n = 58 for questions applying to secondary care, n = 112 for questions applying to midwives, n = 82 for questions applying to primary care midwives.

**Table 2. Participant characteristics, stratified by profession (total n = 643).**

| Variables | Total N | Total n (%) or mean [SD] | Obstetrician/ obstetric resident, n (%) or mean [SD] | Hospital-based midwife, n (%) or mean [SD] | Primary care midwife, n (%) or mean [SD] |
|---|---|---|---|---|---|
| Female gender | 505 | 463 (92) | 115 (74) | 91 (100) | 257 (99) |
| *Prefer not to answer* | | 8 (2) | 6 (4) | 0 (0) | 2 (1) |
| Age in years, mean [SD] | 504 | 41 [11] | 44 [9] | 42 [7] | 38 [11] |
| Work experience in years, mean [SD] | 504 | 16 [10] | 17 [9] | 17 [9] | 15 [10] |
| Country of education abroad | 504 | 75 (15) | 3 (2) | 33 (37) | 39 (15) |
| Employment | 504 | | | | |
| *Paid employment* | | 194 (38) | 85 (55) | 88 (98) | 21 (8) |
| *Self-employed without partnership* | | 99 (20) | 3 (2) | 0 (0) | 96 (37) |
| *Partnership* | | 202 (40) | 64 (41) | 1 (1) | 137 (53) |
| *Other* | | 9 (2) | 3 (2) | 1 (1) | 5 (2) |
| Region of work (consortia) | 495 | | | | |
| *North Netherlands* | | 68 (14) | 18 (12) | 12 (14) | 38 (15) |
| *Northwest Netherlands* | | 86 (17) | 32 (21) | 17 (19) | 37 (14) |
| *Central Netherlands* | | 72 (15) | 20 (13) | 5 (6) | 47 (18) |
| *East Netherlands* | | 71 (14) | 16 (11) | 12 (14) | 43 (17) |
| *Southwest Netherlands* | | 127 (26) | 40 (26) | 27 (31) | 60 (23) |
| *Brabant* | | 25 (10) | 19 (13) | 11 (13) | 25 (10) |
| *Limburg* | | 16 (3) | 6 (4) | 4 (5) | 6 (2) |
| Knowledge about scientific foundation of the guideline[a], mean [SD] | 500 | 3.8 [0.9] | 4.3 [0.7] | 3.8 [0.9] | 3.5 [0.9] |
| *Unsure (n (%))* | | 12 (2) | 4 (3) | 2 (2) | 6 (2) |
| Positive attitude towards guidelines in general*, mean [SD] | 500 | 3.7 [0.8] | 4.1 [0.6] | 4.0 [0.7] | 3.4 [0.8] |
| *Unsure (n (%))* | | 10 (2) | 3 (2) | 2 (2) | 5 (2) |
| Experience of capacity problem*, mean [SD] | 502 | 3.6 [0.8] | 3.6 [0.8] | 3.6 [0.7] | 3.7 [0.8] |
| *Unsure (n (%))* | | 1 (0) | 0 (0) | 0 (0) | 1 (0) |
| Birth Beliefs Scale | | | | | |
| *Medical view, mean [SD]* | 496 | 2.8 [0.6] | 3.3 [0.4] | 2.9 [0.4] | 2.4 [0.5] |
| *Natural view, mean [SD]* | 496 | 3.9 [0.6] | 3.3 [0.5] | 3.7 [0.4] | 4.2 [0.5] |
| Aware of existence of guideline | 643 | | | | |
| *Module 1* | 643 | 594 (92) | 179 (98) | 105 (88) | 310 (91) |
| *Module 2* | 643 | 435 (68) | 164 (90) | 75 (63) | 196 (57) |
| *If aware of existence:* | | | | | |
| Read further than recommendations | | | | | |
| *Module 1* | 594 | 390 (66) | 134 (75) | 68 (65) | 188 (61) |
| *Module 2* | 435 | 262 (60) | 115 (70) | 47 (63) | 100 (51) |
| Did not read guideline | | | | | |
| *Module 1* | 594 | 28 (5) | 2 (1) | 6 (6) | 20 (6) |
| *Module 2* | 435 | 27 (6) | 3 (2) | 6 (8) | 18 (9) |

[a] 5-point Likert scale.

Number of missings: n = 138 for gender, n = 139 for age, working experience, country of education, and employment, n = 148 for region of work, n = 143 for knowledge about scientific foundation of the guideline, and positive attitude towards guidelines in general, n = 141 for experience of capacity problem, n = 147 for Birth Beliefs Scale, n = 0 for aware of existence of guideline, read further than recommendations, and did not read guideline (these questions were asked at the beginning of the survey, while the others were asked at the end).

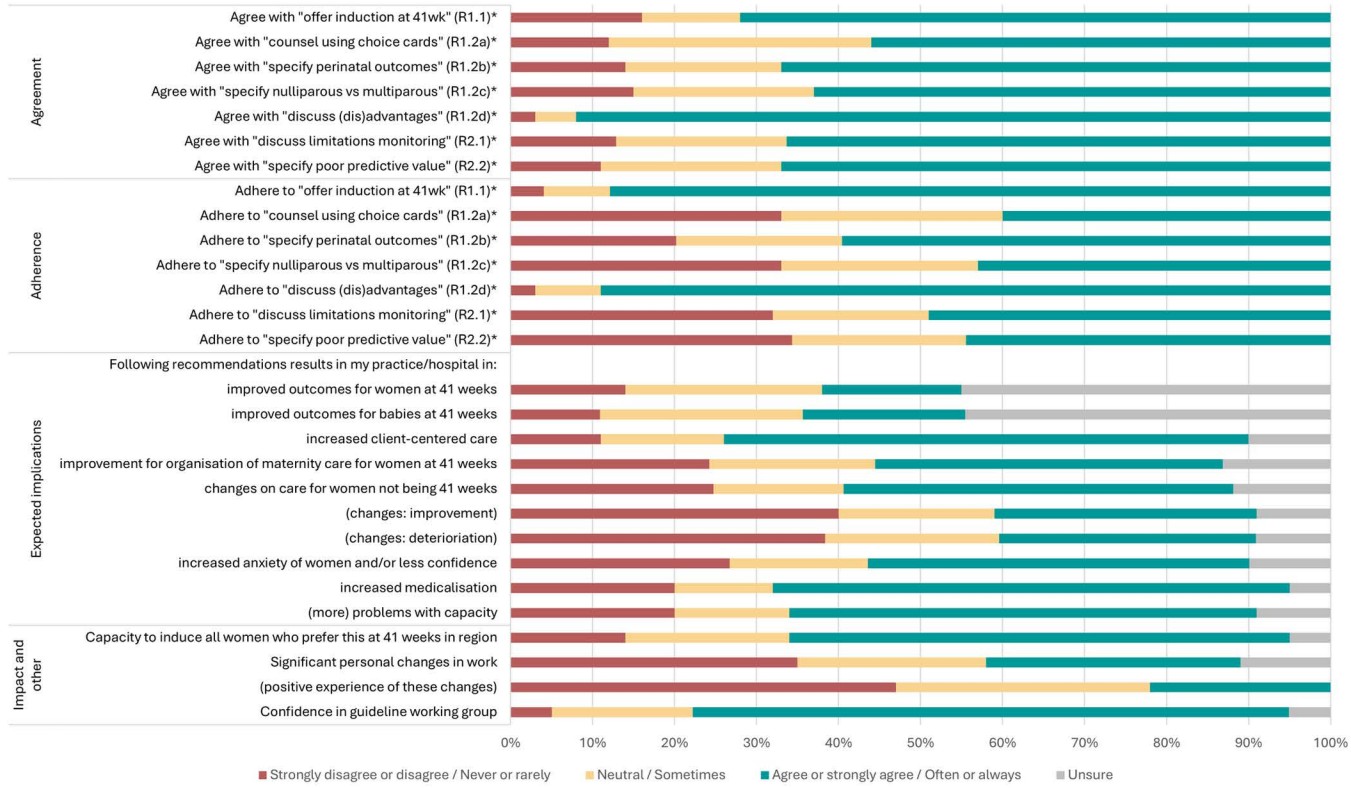

**Fig 1a. Agreement, adherence, opinions, and experiences of care providers on late-term management, rated with a 5-point Likert Scale (n=643). Legend:** 5-point Likert Scales were used. Standard Likert Scale was used for questions on agreement, expected implications and impact and other (except the question on capacity). Frequency Likert Scale was used for questions on adherence and the question on capacity. Standard Likert Scale options: strongly disagree – disagree– neutral –agree – strongly agree – (unsure)Frequency Likert Scale options: never – rarely – sometimes – often – always – (unsure) * Recommendations in the guideline: R1.1: Offer the pregnant woman with a singleton in cephalic presentation the option of inducing labour from 41 weeks (287 days amenorrhea) alongside the option to wait. R1.2a: Counsel using the choice cards associated with the module. R1.2b: Specify the effects on perinatal outcomes such as perinatal mortality and NICU admissions. R1.2c: Specify the potential increased risk for nulliparous compared to multiparous women. R1.2d: In this context, the advantages and disadvantages of both induction and expectant management should be discussed. R2.1: Discuss with the pregnant woman the limitations of fetal monitoring at 41 weeks if she wishes to wait. R2.2: Specify that the cardiotocography (CTG) and ultrasound, in the context of expectant management beyond 41 weeks, do not have a good predictive value for pregnancy outcomes.

multiparous women (R1.2c). The lowest level of agreement (56%) was found for the recommendation to perform shared decision-making using the choice cards accompanying the guideline (R1.2a).

## Implications, impact and other

There was considerable uncertainty among healthcare providers regarding the benefits to mothers and neonates of adhering to the recommendations in practice (45%; Fig 1a). Only 17% believed that the recommendations would improve outcomes for women at 41 weeks and 20% for neonates at 41 weeks. In general, primary care midwives were more negative about the impact, hospital-based midwives were slightly more positive, and obstetricians were the most positive (Fig 1b). Most participants saw benefits for client-centred care (64%), and 42% saw organizational benefits. However, 47% reported increased anxiety and/or less confidence among women, with 63% indicating a rise in medicalization and 57% in capacity issues. This opinion was especially prominent among primary care midwives and less among obstetricians. Despite this, the majority felt that regional capacity was adequate (61%). Primary care midwives were more likely to

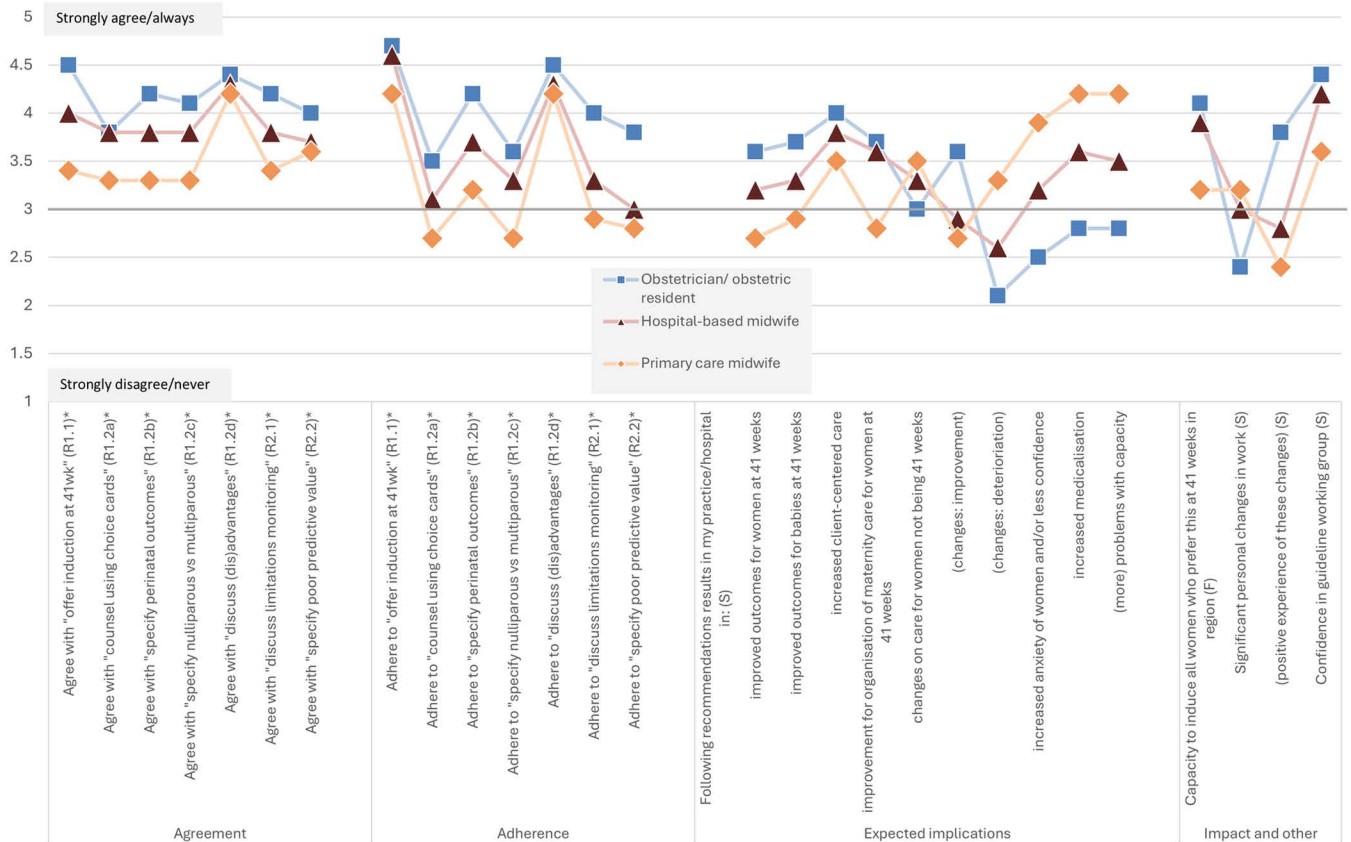

**Fig 1b. Agreement, adherence, opinions, and experiences of care providers on late-term management (mean score on a 5-point Likert Scale), stratified by profession (n=643). Legend:** 5-point Likert Scales were used. Standard Likert Scale was used for questions on agreement, expected implications and impact and other (except the question on capacity). Frequency Likert Scale was used for questions on adherence and the question on capacity. Standard Likert Scale options: strongly disagree (1) – disagree (2) – neutral (3) – agree (4) – strongly agree (5) – (unsure) Frequency Likert Scale options: never (1) – rarely (2) – sometimes (3) – often (4) – always (5) – (unsure) * Recommendations in the guideline: R1.1: Offer the pregnant woman with a singleton in cephalic presentation the option of inducing labour from 41 weeks (287 days amenorrhea) alongside the option to wait. R1.2a: Counsel using the choice cards associated with the module. R1.2b: Specify the effects on perinatal outcomes such as perinatal mortality and NICU admissions. R1.2c: Specify the potential increased risk for nulliparous compared to multiparous women. R1.2d: In this context, the advantages and disadvantages of both induction and expectant management should be discussed. R2.1: Discuss with the pregnant woman the limitations of fetal monitoring at 41 weeks if she wishes to wait. R2.2: Specify that the cardiotocography (CTG) and ultrasound, in the context of expectant management beyond 41 weeks, do not have a good predictive value for pregnancy outcomes.

perceive capacity issues, both generally and within their regions. Almost half of the participants (48%) reported changes in the care for women not being 41 weeks, with the majority of midwives observing deterioration and the majority of obstetricians observing improvement. Of the 31% who experienced personal changes, most found these to be negative (47%), with the highest rates among midwives. Confidence in the guideline working group was high across all professions (72%).

## Associations

The highest associations with agreement with the recommendations were: profession, BBS, and attitude towards guidelines (Table 4). For adherence to the recommendations, the highest associations were profession, attitude towards guidelines, and awareness and reading of the guideline (Table 5). Overall, obstetricians and hospital-based midwives were more likely to agree with and adhere to the recommendations compared to primary care midwives, particularly

**Table 3.** Preferred policy option of care providers who disagreed with offering induction at 41 weeks, stratified by profession (n = 172).

| Variables | Total n (%)ᵃ | Obstetrician/ obstetric resident, n (%) | Hospital-based midwife, n (%) | Primary care midwife, n (%) |
|---|---|---|---|---|
| Total | **172** | **10 (6)** | **17 (10)** | **145 (84)** |
| Advice for induction at 41 + 0 weeks to all women | 5 (3) | 3 (30) | 1 (6) | 1 (1) |
| Offering induction at other time point<br>• 39 + 0<br>• 40 + 0<br>• 41 + 3<br>• 41 + 5 | 25 (15) | 2 (20) | 1 (6) | 22 (15) |
| Expectant management until 42 + 0 weeks | 121 (70) | 1 (10) | 7 (41) | 113 (78) |
| Expectant management without boundary | 25 (15) | 1 (10) | 1 (6) | 23 (16) |
| Offering induction at 41 + 0 weeks only to nulliparous women. | 20 (12) | 0 (0) | 3 (18) | 17 (12) |
| All women offered induction at 41 weeks, but advice to nulliparous women | 10 (6) | 4 (40) | 3 (18) | 3 (2) |
| Other | 11 (6) | 0 (0) | 3 (18) | 8 (6) |

ᵃTotal number can exceed 100%, because more than one answer option was possible.

Adherence to the guideline was generally lower among primary care midwives than the other professional groups, except discussing the (dis)advantages of IOL and EM. While 16% of all participants disagreed with recommendation 1.1 "(offer IOL at 41 weeks"), the majority often or always adhered to it (87%). In contrast, recommendations 1.2a ("counsel using choice cards") and 1.2c ("specify nulliparous vs multiparous") and recommendations in module 2 on monitoring had higher levels of agreement but much lower adherence rates. Module 2 was notably underutilized with adherence rates of 49% for R2.1 ("discuss limitations of monitoring") and 44% for R2.2 ("specify poor predictive value") for the total population.

regarding the recommendations on offering IOL and specifying adverse perinatal outcomes. A positive attitude towards guidelines was associated with higher agreement and adherence rates. A medical view on the BBS was associated with more agreement, regardless of the profession as adjustments were applied for profession. A natural view was associated with lower agreement of R1.1 ("offer IOL at 41 weeks") and a medical view was associated with a higher adherence to R1.1. Although care providers who were well-acquainted with and had thoroughly read the guidelines, were more likely to adhere to the recommendations, this barely influenced the agreement with the guideline. Reading the guideline was only associated with a higher agreement for "specify nulliparous versus multiparous". Regional differences were minimal.

## Discussion

### Main findings

In summary, despite considerable uncertainty among care providers about the positive outcomes of the recommendations for women and babies, most care providers agreed with offering IOL at 41 weeks alongside the option of EM, and the vast majority adhered to this recommendation. However, the implementation of specific aspects of the guideline in daily practice, such as the use of choice cards, detailed discussion of perinatal outcomes, differentiation according to parity, and limitations of monitoring, was notably lower among all care providers. The profession of the care provider was the most influential factor in determining agreement with and adherence to the recommendations. Overall, primary care midwives expressed less favourable views toward the guideline and its perceived impact, resulting in a lower rate of adherence. Hospital-based midwives followed a similar trend but to a lesser extent. In contrast, obstetricians had the most positive attitudes and the highest rate of guideline adherence.

Additionally, beliefs regarding natural versus medical beliefs about childbirth influenced agreement with the recommendations, regardless of professional background. Finally, thorough reading of the guideline was positively associated with adherence to the guideline.

**Table 4. Association between participant characteristics and agreement on recommendations in the guideline (n = 495).**

| Variables | Regression coefficient (β) | | | | |
|---|---|---|---|---|---|
| | Agree with "offer induction at 41wk" (R1.1)[a] | Agree with "counsel using choice cards" (R1.2a)[a] | Agree with "specify perinatal outcomes" (R1.2b)[a] | Agree with "specify nulliparous vs multiparous" (R1.2c)[a] | Agree with "discuss (dis)advantages" (R1.2d)[a] |
| | adj[b] coefficient (95% CI) | adj[b] coefficient (95% CI) | adj[b] coefficient (95% CI) | adj[b] coefficient (95% CI) | adj[b] coefficient (95% CI) |
| Profession | | | | | |
| *Obstetrician/ obstetric resident* | **1.00 (0.77-1.24)** | **0.60 (0.40-0.80)** | **0.96 (0.76-1.16)** | **0.78 (0.58-0.98)** | **0.36 (0.20-0.53)** |
| *Hospital-based midwife* | **0.62 (0.36-0.88)** | **0.34 (0.12-0.56)** | **0.54 (0.32-0.76)** | **0.52 (0.30-0.75)** | **0.21 (0.03-0.40)** |
| *Primary care midwife* | ref | ref | ref | ref | ref |
| Male gender | 0.24 (−0.17-0.64) | −0.22 (−0.56-0.13) | 0.29 (−0.05-0.64) | 0.07 (−0.28-0.41) | 0.02 (−0.27-0.30) |
| Age in years | −0.01 (−0.3-0.01) | **−0.03 (−0.04-−0.01)** | **−0.02 (−0.04-−0.01)** | 0.00 (−0.02-0.01) | **−0.03 (−0.04-−0.02)** |
| Working experience in years | 0.00 (−0.02-0.02) | 0.02 (−0.002-0.03) | 0.01 (−0.004-0.03) | 0.00 (−0.01-0.02) | 0.01 (−0.002-0.03) |
| Country of education abroad | −0.07 (−0.35-0.20) | −0.02 (−0.26-0.21) | 0.05 (−0.18-0.29) | −0.07 (−0.31-0.16) | −0.05 (−0.25-0.14) |
| Region of work (consortia)[c] | | | | | |
| *North Netherlands* | 0.05 (−0.19-0.28) | **−0.23 (−0.44-−0.03)** | 0.02 (−0.18-0.22) | −0.14 (−0.34-0.06) | −0.13 (−0.30-0.04) |
| *Northwest Netherlands* | 0.09 (−0.13-0.30) | −0.08 (−0.27-0.10) | 0.05 (−0.13-0.24) | 0.00 (−0.19-0.18) | −0.10 (−0.25-0.06) |
| *Central Netherlands* | −0.06 (−0.30-0.17) | −0.02 (−0.22-0.18) | −0.11 (−0.31-0.09) | **−0.21 (−0.40-−0.01)** | −0.07 (−0.24-0.10) |
| *East Netherlands* | −0.11 (−0.35-0.12) | 0.07 (−0.13-0.27) | −0.05 (−0.25-0.15) | 0.01 (−0.18-0.21) | 0.11 (−0.06-0.28) |
| *Southwest Netherlands* | −0.11 (−0.30-0.08) | −0.16 (−0.32-0.001) | −0.08 (−0.24-0.09) | 0.07 (−0.09-0.22) | 0.02 (−0.12-0.16) |
| *Brabant* | 0.09 (−0.16-0.35) | 0.07 (−0.15-0.29) | 0.04 (−0.18-0.26) | −0.03 (−0.24-0.19) | −0.04 (−0.23-0.14) |
| *Limburg* | 0.06 (−0.38-0.50) | 0.36 (−0.03-0.74) | 0.12 (−0.26-0.50) | 0.30 (−0.08-0.67) | 0.21 (−0.11-0.54) |
| Knowledge about scientific foundation of the guideline | 0.09 (−0.02-0.20) | 0.06 (−0.04-0.15) | 0.06 (−0.03-0.16) | 0.04 (−0.05-0.13) | 0.07 (−0.003-0.15) |
| Positive attitude towards guidelines in general | **0.30 (0.17-0.42)** | **0.27 (0.16-0.38)** | **0.24 (0.13-0.35)** | **0.16 (0.05-0.27)** | **0.15 (0.06-0.24)** |
| Experience of capacity problem | **−0.13 (−0.26-−0.01)** | −0.05 (−0.16-0.05) | −0.09 (−0.20-0.01) | −0.09 (−0.20-0.01) | −0.04 (−0.13-0.04) |
| Birth Beliefs Scale | | | | | |
| *Medical view, mean (p5-p95)* | **0.40 (0.20-0.60)** | **0.24 (0.06-0.41)** | **0.20 (0.03-0.38)** | **0.21 (0.04-0.39)** | 0.06 (−0.08-0.21) |
| *Natural view, mean (p5-p95)* | **−0.23 (−0.43-−0.03)** | 0.02 (−0.15-0.20) | −0.10 (−0.27-0.07) | −0.14 (−0.31-0.03) | 0.10 (−0.05-0.24) |
| Aware of existence of module 1 | 0.14 (−0.2-0.50) | 0.21 (−0.10-0.52) | −0.04 (−0.35-0.27) | 0.17 (−0.13-0.48) | 0.12 (−0.13-0.38) |

*(Continued)*

**Table 4.** (Continued)

| Variables | Regression coefficient (β) | | | | |
| --- | --- | --- | --- | --- | --- |
| | Agree with "offer induction at 41wk" (R1.1)[a] | Agree with "counsel using choice cards" (R1.2a)[a] | Agree with "specify perinatal outcomes" (R1.2b)[a] | Agree with "specify nulliparous vs multiparous" (R1.2c)[a] | Agree with "discuss (dis) advantages" (R1.2d)[a] |
| | adj[b] coefficient (95% CI) | adj[b] coefficient (95% CI) | adj[b] coefficient (95% CI) | adj[b] coefficient (95% CI) | adj[b] coefficient (95% CI) |
| Read further than recommendations of module 1 | 0.03 (−0.17-0.23) | 0.13 (−0.04-0.30) | 0.10 (−0.07-0.28) | **0.27 (0.10-0.44)** | 0.04 (−0.10-0.19) |
| Did not read module 1 | −0.05 (−0.50-0.39) | −0.03 (−0.41-0.34) | −0.17 (−0.55-0.21) | −0.24 (−0.61-0.14) | 0.01 (−0.30-0.32) |

**Legenda:**

[a]Recommendations in the guideline, included in this table:

R1.1:Offer the pregnant woman with a singleton in cephalic presentation the option of inducing labour from 41 weeks (287 days amenorrhea) alongside the option to wait.

R1.2a:Counsel using the choice cards associated with the module.

R1.2b:Specify the effects on perinatal outcomes such as perinatal mortality and NICU admissions.

R1.2c:Specify the potential increased risk for nulliparous compared to multiparous women.

R1.2d:In this context, the advantages and disadvantages of both induction and expectant management should be discussed.

[b]adjusted regression coefficients of Likert scales in multivariable analyses. Adjusted for profession, gender, age, working experience, country of education, and region of work (with coefficient of zero meaning no difference in Likert scale).

[c]The association between region of work and outcomes was only adjusted for profession. The reference category is the weighted mean.

Regression coefficients in bold type are statistically significant. A regression coefficient indicates the increase in the Likert scale of the outcome if the dependent variable increases with 1 unit.

## Interpretation

In this study, we examined adoption, fidelity, and penetration from the perspective of maternity care providers, which we believed would offer valuable insights into how the guidelines are perceived and implemented in practice. We observed considerable variation in the implementation of the different recommendations. While adherence to offering IOL at 41 weeks alongside the option of EM was high, adherence to recommendations concerning specific aspects of the shared decision-making conversation was much lower. Although this research was conducted in the Netherlands, the results are also relevant for other countries. Previous literature has identified various factors associated with guideline adherence [25]. The discrepancy in adherence in our study may be due to several of these factors, including 'knowledge'—care providers' awareness of the guideline's existence and content—and 'skills'—their understanding and training in the guideline's content [25]. This aligns with our findings, which showed greater adherence when the guideline was known, understood, and read. Obstetricians scored higher on these factors, had generally more positive attitudes towards guidelines, and scored higher on adherence. This suggests that awareness, comprehension, and a positive attitude towards guidelines are factors that influence the degree to which guidelines are adhered to.

Differences in adherence to the recommendations between obstetricians, hospital-based midwives, and primary care midwives may also be explained by their different roles and beliefs about birth. Primary care midwives typically care for low-risk women throughout the entire pregnancy, encounter fewer complications, and relinquish care when IOL is performed. Additionally, primary care midwives scored higher in their preference for a natural view of pregnancy and birth, which may affect their agreement with and adherence to specific recommendations, such as specifying adverse perinatal outcomes in shared decision-making conversations. They also more frequently disagreed with statements regarding

**Table 5. Association between participant characteristics and adherence to the guideline (n = 483).**

| | Regression coefficient (β) | | | | |
|---|---|---|---|---|---|
| **Variables** | Adherence to "offer induction at 41wk" (R1.1)a | Adherence to "counsel using choice cards" (R1.2a)a | Adherence to "specify perinatal outcomes" (R1.2b)a | Adherence to "specify nulliparous vs multiparous" (R1.2c)a | Adherence to "discuss (dis)advantages" (R1.2d)a |
| | adjb coefficient (95% CI) | adjb coefficient (95% CI) | adjb coefficient (95% CI) | adjb coefficient (95% CI) | adjb coefficient (95% CI) |
| Profession | | | | | |
| *Obstetrician/ obstetric resident* | **0.50 (0.31-0.69)** | **0.72 (0.44-0.999)** | **0.99 (0.74-1.24)** | **0.79 (0.50-1.08)** | **0.39 (0.21-0.57)** |
| *Hospital-based midwife* | **0.31 (0.09-0.53)** | 0.31 (−0.01-0.63) | **0.50 (0.21-0.79)** | **0.50 (0.18-0.83)** | 0.13 (−0.08-0.33) |
| *Primary care midwife* | ref | ref | ref | ref | ref |
| Male gender | 0.06 (−0.26-0.39) | −0.16 (−0.64-0.32) | 0.08 (−0.35-0.52) | 0.31 (−0.18-0.80) | −0.05 (−0.35-0.26) |
| Age in years | **−0.02 (−0.03-−0.003)** | 0.00 (−0.03-0.02) | **−0.02 (−0.04-−0.002)** | 0.00 (−0.02-0.03) | **−0.04 (−0.06-−0.03)** |
| Working experience in years | 0.01 (−0.003-0.03) | 0.00 (−0.03-0.02) | 0.02 (−0.004-0.04) | 0.02 (−0.01-0.04) | **0.03 (0.02-0.05)** |
| Country of education abroad | 0.08 (−0.15-0.30) | −0.12 (−0.45-0.21) | −0.20 (−0.50-0.09) | −0.23 (−0.56-0.11) | −0.12 (−0.33-0.09) |
| Region of work (consortia)c | | | | | |
| *North Netherlands* | 0.04 (−0.15-0.23) | **−0.47 (−0.75-−0.19)** | −0.14 (−0.40-0.11) | −0.21 (−0.50-0.08) | −0.13 (−0.31-0.06) |
| *Northwest Netherlands* | 0.03 (−0.15-0.21) | −0.20 (−0.46-0.06) | 0.00 (−0.24-0.24) | 0.02 (−0.25-0.29) | −0.08 (−0.25-0.10) |
| *Central Netherlands* | **−0.24 (−0.43- −0.06)** | 0.14 (−0.14-0.41) | −0.16 (−0.41-0.09) | −0.16 (−0.45-0.12) | 0.00 (−0.18-0.19) |
| *East Netherlands* | −0.03 (−0.22-0.16) | −0.02 (−0.30-0.26) | −0.06 (−0.31-0.19) | 0.04 (−0.25-0.33) | 0.00 (−0.19-0.18) |
| *Southwest Netherlands* | −0.02 (−0.17-0.14) | 0.02 (−0.21-0.24) | −0.04 (−0.25-0.16) | 0.23 (−0.01-0.46) | 0.01 (−0.14-0.16) |
| *Brabant* | −0.08 (−0.29-0.12) | 0.19 (−0.12-0.49) | 0.12 (−0.15-0.40) | −0.25 (−0.56-0.07) | 0.15 (−0.05-0.35) |
| *Limburg* | 0.30 (−0.06-0.66) | 0.35 (−0.18-0.87) | 0.29 (−0.19-0.76) | 0.34 (−0.21-0.88) | 0.05 (−0.30-0.39) |
| Knowledge about scientific foundation of the guideline | 0.06 (−0.03-0.15) | **0.18 (0.04-0.31)** | **0.19 (0.07-0.31)** | **0.21 (0.08-0.35)** | 0.05 (−0.04-0.13) |
| Positive attitude towards guidelines in general | **0.21 (0.10-0.31)** | **0.30 (0.14-0.45)** | **0.16 (0.02-0.30)** | **0.22 (0.06-0.37)** | 0.10 (−0.004-0.20) |
| Experience of capacity problem | 0.05 (−0.05-0.15) | **−0.19 (−0.34-−0.04)** | 0.00 (−0.13-0.14) | −0.03 (−0.19-0.12) | 0.06 (−0.03-0.16) |
| Birth Beliefs Scale | | | | | |
| *Medical view, mean (p5-p95)* | **0.31 (0.15-0.48)** | 0.24 (−0.002-0.49) | −0.04 (−0.26-0.18) | 0.15 (−0.11-0.40) | −0.12 (−0.28-0.04) |
| *Natural view, mean (p5-p95)* | −0.09 (−0.25-0.08) | −0.12 (−0.36-0.13) | 0.09 (−0.14-0.31) | 0.02 (−0.24-0.27) | 0.10 (−0.06-0.25) |
| Aware of existence of module 1 | 0.04 (−0.26-0.33) | **1.14 (0.72-1.57)** | **0.41 (0.02-0.80)** | **0.88 (0.44-1.32)** | 0.28 (−0.003-0.55) |

*(Continued)*

| Variables | Regression coefficient (β) | | | | |
| --- | --- | --- | --- | --- | --- |
| | Adherence to "offer induction at 41wk" (R1.1)a | Adherence to "counsel using choice cards" (R1.2a)a | Adherence to "specify perinatal outcomes" (R1.2b)a | Adherence to "specify nulliparous vs multiparous" (R1.2c)a | Adherence to "discuss (dis) advantages" (R1.2d)a |
| | adjb coefficient (95% CI) | adjb coefficient (95% CI) | adjb coefficient (95% CI) | adjb coefficient (95% CI) | adjb coefficient (95% CI) |
| Read further than recommendations of module 1 | **0.20 (0.03-0.36)** | **0.66 (0.43-0.89)** | **0.47 (0.26-0.69)** | **0.71 (0.48-0.95)** | **0.23 (0.08-0.38)** |
| Did not read module 1 | −0.16 (−0.54-0.22) | **−0.89 (−1.43-−0.35)** | **−0.92 (−1.41- −0.43)** | **−0.86 (−1.42-−0.29)** | **−0.74 (−1.09-−0.40)** |

**Legenda:**

aRecommendations in the guideline, included in this table:

R1.1:Offer the pregnant woman with a singleton in cephalic presentation the option of inducing labour from 41 weeks (287 days amenorrhea) alongside the option to wait.

R1.2a:Counsel using the choice cards associated with the module.

R1.2b:Specify the effects on perinatal outcomes such as perinatal mortality and NICU admissions.

R1.2c:Specify the potential increased risk for nulliparous compared to multiparous women.

R1.2d:In this context, the advantages and disadvantages of both induction and expectant management should be discussed.

badjusted regression coefficients of Likert scales in multivariable analyses. Adjusted for profession, gender, age, working experience, country of education, and region of work (with coefficient of zero meaning no difference in Likert scale).

cThe association between region of work and outcomes was only adjusted for profession. The reference category is the weighted mean.

Regression coefficients in bold type are statistically significant. A regression coefficient indicates the increase in the Likert scale of the outcome if the dependent variable increases with 1 unit.

the positive impact of the guideline on maternal or perinatal outcomes and the organization of care. This aligns with the findings of Coates et al. (2021) who reported that midwives generally favour a physiological approach to birth, whereas medical staff are more inclined to support IOL [26,27]. We also observed that the perspective of hospital-based midwives were generally between those of primary care midwives and obstetricians. Although they are midwives, working in a more medicalised setting is likely to influence their views. However, although professions and views are strongly correlated, the two are not interchangeable, because both the natural and the medical view exist in all professions. After adjusting for profession, a medical view was associated with higher agreement with the recommendations, suggesting that the guideline is more medically oriented, leading to more frequent disagreement between care providers with differing views.

Although previous literature [25] has not emphasized that the nature of specific recommendations may influence adherence, our study suggests that this is indeed the case. While the recommendation to offer choices is straightforward to implement, adherence to more detailed recommendations on shared decision-making is more challenging. Both aspects are covered in the guideline, yet care providers may prioritize visible elements, such as offering choices, possibly also to avoid feedback from pregnant women and colleagues about non-compliance. In contrast, the specific details of these conversations are less visible, making non-adherence less likely to be noticed. Furthermore, the adherence to recommendations of Module 2 were low, which may suggest that these were perceived as somewhat unrealistic.

Although almost all participants agreed that the advantages and disadvantages of the choice options should be discussed, only 67% supported specifying perinatal outcomes. Literature on risk communication indicates that presenting numerical data enhances understanding of probabilities and avoids overestimation of risk, compared to using qualitative

descriptors such as 'higher' or 'lower' [28]. Therefore, specifying the perinatal outcomes for each choice option and using choice cards, as recommended in the guideline, is advised to improve clarity and support informed decision-making.

Furthermore, 45% of all participants expressed uncertainty about whether this guideline results in improved maternal or perinatal outcomes. This may be due to ambivalent results from the literature. Several studies show improved outcomes for IOL at 41 weeks compared to EM [17,18], while others report less favourable outcomes [29–31], particularly regarding caesarean section rates [30,32]. Although randomized controlled trials (RCTs) are considered the most effective design for assessing the effectiveness of an intervention, the implementation of a new policy at the national level may yield results that differ from what is expected from RCTs [32,33]. It is therefore recommended that the shift in late-term induction policy be further evaluated to gain more insight into the actual impact of such a national policy change and to provide care providers with greater certainty about the most optimal policy. Additionally, an analysis of the cost implications of implementing a more restrictive IOL policy is needed, as the cost-effectiveness analysis of the INDEX trial showed that IOL may be cost-effective for nulliparous women but is unlikely to be cost-effective for multiparous women.

### Strengths and limitations

This study included a large sample size with good representation from all professions, and the comprehensive nature of the questionnaire provided valuable insights into the perspectives of care providers (11.6% of all working obstetricians and 11.2% of midwives). Although the questionnaire was not validated, interviews were conducted to ensure the validity of the measures. Additionally, the Birth Beliefs Scale, a validated tool, was included.

There may have been some degree of the Hawthorne effect, as participation in the survey may have increased awareness of the guideline's existence. Furthermore, selection bias may have occurred, first because one region was underrepresented, and secondly, as participants were likely to be more interested in the content of the guideline than the average care provider. However, it remains unclear whether this bias specifically affected more critical care providers or those who feel more positively about the guideline.

### Conclusions

Most care providers agreed with and adhered to the recommendation to offer IOL at 41 weeks alongside the option of EM, but fewer care providers adhered to the recommendation on specific details regarding the shared-decision making conversation. Among care providers, there was uncertainty about the positive implications of the guideline. Further research is needed to evaluate maternal and neonatal outcomes after the implementation of the guideline.

### Supporting information

**S1 File. Questionnaire LATE-study.**
(DOCX)

### Acknowledgments

We thank Het Buikencollectief (a women representative organisation) for their constructive collaboration and Rik van Eekelen for his statistical support. We also thank Suzanne Sturkenboom, Marieke Mink, Lauren Ancion and Liesbeth van Esseveldt for their effort and help.

### Author contributions

**Conceptualization:** Anna E. Seijmonsbergen-Schermers, Fatima Hammiche, Judit K.J. Keulen, Caroline J. Bax, Lilian L. Peters, Corine J. Verhoeven.

**Data curation:** Anna E. Seijmonsbergen-Schermers, Fatima Hammiche.

**Formal analysis:** Anna E. Seijmonsbergen-Schermers.

**Funding acquisition:** Anna E. Seijmonsbergen-Schermers.

**Investigation:** Anna E. Seijmonsbergen-Schermers.

**Methodology:** Anna E. Seijmonsbergen-Schermers, Fatima Hammiche, Hannah de Klerk, Judit K.J. Keulen, Caroline J. Bax, Lilian L. Peters, Corine J. Verhoeven.

**Project administration:** Anna E. Seijmonsbergen-Schermers.

**Supervision:** Corine J. Verhoeven.

**Visualization:** Anna E. Seijmonsbergen-Schermers.

**Writing – original draft:** Anna E. Seijmonsbergen-Schermers.

**Writing – review & editing:** Fatima Hammiche, Hannah de Klerk, Judit K.J. Keulen, Caroline J. Bax, Lilian L. Peters, Corine J. Verhoeven.

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
