## [Decision Letter · Decision Letter 0]

15 Apr 2025

PONE-D-25-07825Maternity care providers’ perspectives on late-term gestation management (LATE-study): a cross-sectional survey.PLOS ONE

Dear Dr. Seijmonsbergen-Schermers,

Thank you for submitting your manuscript to PLOS ONE. After careful consideration, we feel that it has merit but does not fully meet PLOS ONE’s publication criteria as it currently stands. Therefore, we invite you to submit a revised version of the manuscript that addresses the points raised during the review process.

Kindly respond to the review comments and in particular address the concern raised by reviewer 2 about possible salami slicing.

We strongly discourage the unnecessary division of related work into separate manuscripts, and we will not consider manuscripts that are divided into parts. Each submission to PLOS One must be written as an independent unit and should not rely on any work that has not already been accepted for publication. Please upload a copy of the related submission entitled "Shared decision-making on late-term gestation management (LATE-study): a cross-sectional survey among maternity care providers." as an 'other' file when you resubmit your manuscript.

We look forward to receiving your revised manuscript.

Kind regards,

Patrick Ifeanyi Okonta, MBBCh, MPH, FWACS, FMCOG, MD, DRH

Academic Editor

PLOS ONE

Journal Requirements:

I have read the journal's policy and the authors of this manuscript have the following competing interests:

Hannah de Klerk (midwife) and Caroline Bax (obstetrician) were members of the working group who developed the Dutch national guideline on management of pregnancies at 41 weeks’ gestation onwards.

Please respond by return email with your amended Competing Interests Statement and we will change the online submission form on your behalf.

4. Please note that your Data Availability Statement is currently missing [the repository name and/or the DOI/accession number of each dataset OR a direct link to access each database]. If your manuscript is accepted for publication, you will be asked to provide these details on a very short timeline. We therefore suggest that you provide this information now, though we will not hold up the peer review process if you are unable.

5. Please amend your list of authors on the manuscript to ensure that each author is linked to an affiliation. Authors’ affiliations should reflect the institution where the work was done (if authors moved subsequently, you can also list the new affiliation stating “current affiliation:….” as necessary).

6. Please amend either the abstract on the online submission form (via Edit Submission) or the abstract in the manuscript so that they are identical.

7. We note that you have included the phrase “data not shown” in your manuscript. Unfortunately, this does not meet our data sharing requirements. PLOS does not permit references to inaccessible data. We require that authors provide all relevant data within the paper, Supporting Information files, or in an acceptable, public repository. Please add a citation to support this phrase or upload the data that corresponds with these findings to a stable repository (such as Figshare or Dryad) and provide and URLs, DOIs, or accession numbers that may be used to access these data. Or, if the data are not a core part of the research being presented in your study, we ask that you remove the phrase that refers to these data.

8. Please remove all personal information, ensure that the data shared are in accordance with participant consent, and re-upload a fully anonymized data set.

Additional guidance on preparing raw data for publication can be found in our Data Policy (https://journals.plos.org/plosone/s/data-availability#loc-human-research-participant-data-and-other-sensitive-data) and in the following article: http://www.bmj.com/content/340/bmj.c181.long .

Additional Editor Comments :

Dear Authors,

Kindly respond to the review comments and in particular address the concern raised by reviewer 2 about possible salami slicing.

We strongly discourage the unnecessary division of related work into separate manuscripts, and we will not consider manuscripts that are divided into parts. Each submission to PLOS One must be written as an independent unit and should not rely on any work that has not already been accepted for publication. Please upload a copy of the related submission entitled "Shared decision-making on late-term gestation management (LATE-study): a cross-sectional survey among maternity care providers." as an 'other' file when you resubmit your manuscript."

Reviewers' comments:

Reviewer's Responses to Questions

**Comments to the Author**

1. Is the manuscript technically sound, and do the data support the conclusions?

Reviewer #1: Partly

Reviewer #2: Yes

2. Has the statistical analysis been performed appropriately and rigorously? 

Reviewer #1: I Don't Know

Reviewer #2: Yes

3. Have the authors made all data underlying the findings in their manuscript fully available?

Reviewer #1: Yes

Reviewer #2: No

4. Is the manuscript presented in an intelligible fashion and written in standard English?

Reviewer #1: Yes

Reviewer #2: Yes

5. Review Comments to the Author

Reviewer #1: ORIGINALITY: As pointed out in the Introduction to this paper, there has been a long-standing debate about induction of labour in prolonged pregnancy. Four years ago Drife (Best Practice Res Clin Obstet Gynaecol 2021;77: 3-14) summarised the guidelines at that time as follows:

“A systematic analysis in 2020 identified 49 national and international English-language guidelines which mostly recommended induction between 41and 42weeks [Coates et al, Women Birth 2020;33:219-30]. In the Netherlands, where there is no national guideline, a national survey found substantial differences between midwifery-led and obstetrician-led care [Kortekas et al, BMC Pregnancy and Childbirth 2019;19:181]. A Dutch review in 2018 concluded that evidence was lacking for the recommendation to induce labour at 41 weeks instead of 42 weeks and highlighted the need for more studies [Kuelen et al, Midwifery 2018;66:111-8].”

Such studies were already in progress. The SWEPSIS study (Reference 18 of the present paper) had to be stopped early because of increased perinatal mortality in the expectant management group. That study, and the study of Kuelen et al (Ref 17) along with the systematic review by Alkmark et al (Ref 10), led to a Dutch guideline (Ref 16) developed by midwives, obstetricians, paediatricians and client representatives. The present study is an assessment of how that guideline has been received by the different professional groups in the Netherlands.

SCIENTIFIC RELIABILITY: I have some general comments but I feel unable to comment on the statistical aspects of the paper as I am not a statistician.

The total of 643 complete or partial responses seems a satisfactory number for analysis but it would be helpful to know what proportion this represents of the total number of midwives and doctors in the areas covered.

The questionnaire asked participants about agreement with, and adherence to, the guidelines. Adherence was self-reported. Was there any objective measurement of adherence?

The “Birth Beliefs Scale” attempts to compare two “beliefs”: one, that childbirth is potentially pathological, and the other that it is physiological. Both are true, so the "beliefs" overlap. What the Scale actually reflects are entrenched attitudes, which are hard to quantify. I'm concerned about the word “beliefs” because it implies that these attitudes will never change.

Attitudes can be changed by evidence, and the 2021 Dutch guideline was written because new evidence showed that the benefits of induction marginally outweigh the risks. In the present paper the paragraph before “Strengths and limitations” (“Furthermore 45% of participants ..") discusses only the older evidence and mentions “ambivalent results from the literature”. If the authors are questioning the conclusions of references 16, 17 and 18 they should say so. The recent evidence regarding fetal and maternal outcomes is clear, so the paper's final sentence, “Further research is needed to evaluate maternal and neonatal outcomes after the implementation of the guideline”, seems inappropriate.

CLINICAL IMPORTANCE: Guidelines are meant to guide, not dictate, clinical practice, but the increase in obstetric litigation in developed countries has increased their clinical importance. It has also introduced an element of defensive medicine exemplified by “Module 2” of the Dutch guideline. I have not read that guideline but the summary in “Textbox 2” of this paper suggests that its recommendations are totally unrealistic. The present paper is important because it explores the mistrust between obstetricians and midwives in the Netherlands and reveals the reluctance of some midwives to accept high-quality evidence. I think the paper would be improved by discussing these issues rather than focussing on the details of the data.

Reviewer #2: 1. The scope is narrow for global audience

2.The objectives which were experiences, perspective and impact on daily work of maternity care providers based on the updated Dutch national guideline is more of an audit of local practice

3. The outcome of the 2020 systematic review and metanalyses which showed lower incidence of composite adverse perinatal outcome in nulliparous women, has been widely received worldwide and the evidence is superior to the INDEX and SWEPIS studies which the Swedish guideline was based upon.

4. The justification for the study was that the feasibility and acceptability of the guideline was not yet known. This can be better improved upon

5. The study would have more impact if aspects of the implementation of the guidelines were studied, and the gaps identified in terms of adoption, fidelity, penetration and cost etc. addressed.

6. Part of the work has been published, as volunteered by the authors. Could this be a 'salami slicing'

6. PLOS authors have the option to publish the peer review history of their article (what does this mean? ). If published, this will include your full peer review and any attached files.

**Do you want your identity to be public for this peer review?** For information about this choice, including consent withdrawal, please see our Privacy Policy .

Reviewer #1: No

Reviewer #2: **Yes: ** MOHAMMED BUKAR

---

## [Author Response · Author response to Decision Letter 1]

17 Jun 2025

Reviewers' comments:

Reviewer 1:

We would like to thank the reviewer for its valuable comments.

10. ORIGINALITY: As pointed out in the Introduction to this paper, there has been a long-standing debate about induction of labour in prolonged pregnancy. Four years ago Drife (Best Practice Res Clin Obstet Gynaecol 2021;77: 3-14) summarised the guidelines at that time as follows:

“A systematic analysis in 2020 identified 49 national and international English-language guidelines which mostly recommended induction between 41and 42weeks [Coates et al, Women Birth 2020;33:219-30]. In the Netherlands, where there is no national guideline, a national survey found substantial differences between midwifery-led and obstetrician-led care [Kortekas et al, BMC Pregnancy and Childbirth 2019;19:181]. A Dutch review in 2018 concluded that evidence was lacking for the recommendation to induce labour at 41 weeks instead of 42 weeks and highlighted the need for more studies [Kuelen et al, Midwifery 2018;66:111-8].”

Such studies were already in progress. The SWEPSIS study (Reference 18 of the present paper) had to be stopped early because of increased perinatal mortality in the expectant management group. That study, and the study of Kuelen et al (Ref 17) along with the systematic review by Alkmark et al (Ref 10), led to a Dutch guideline (Ref 16) developed by midwives, obstetricians, paediatricians and client representatives. The present study is an assessment of how that guideline has been received by the different professional groups in the Netherlands.

Thank you for pointing out the importance of our manuscript.

11. SCIENTIFIC RELIABILITY: I have some general comments but I feel unable to comment on the statistical aspects of the paper as I am not a statistician.

The total of 643 complete or partial responses seems a satisfactory number for analysis but it would be helpful to know what proportion this represents of the total number of midwives and doctors in the areas covered.

Based on an estimated total of 4,120 practicing midwives and 1,200 obstetricians in the Netherlands, the coverage in our study is 11.2% for midwives and 11.6% for obstetricians. We have now included this information in the discussion section of the manuscript (page 10), as follows:

“…(11.6% of all working obstetricians and 11.2% of midwives).”

12. The questionnaire asked participants about agreement with, and adherence to, the guidelines. Adherence was self-reported. Was there any objective measurement of adherence?

As our study was based on a self-administered questionnaire among maternity care providers, adherence to the guideline was self-reported, and no objective measurement was included. We acknowledge this as a limitation. However, we consider the risk of social desirability or recall bias to be limited in this context. Care providers are generally well aware of whether they offer induction at 41 weeks, whether they use decision aids such as choice cards, and adherence to shared-decision making recommendations. Moreover, observational studies may also introduce bias due to altered behavior when being observed. To minimize response bias, the questionnaire was anonymous; we did not collect names or email addresses, allowing participants to respond honestly and confidentially.

13. The “Birth Beliefs Scale” attempts to compare two “beliefs”: one, that childbirth is potentially pathological, and the other that it is physiological. Both are true, so the "beliefs" overlap. What the Scale actually reflects are entrenched attitudes, which are hard to quantify. I'm concerned about the word “beliefs” because it implies that these attitudes will never change.

Thank you for this thoughtful comment. We understand the reviewer’s concern regarding the term “beliefs” and acknowledge that the constructs measured by the Birth Beliefs Scale may also be interpreted as reflecting underlying attitudes toward childbirth. However, the Birth Beliefs Scale is a previously validated instrument, and as such, we have used the original terminology to maintain consistency with validation study. While we agree that alternative wording might better capture the complexity of the concepts, adapting the scale’s terminology is beyond the scope of our study.

14. Attitudes can be changed by evidence, and the 2021 Dutch guideline was written because new evidence showed that the benefits of induction marginally outweigh the risks. In the present paper the paragraph before “Strengths and limitations” (“Furthermore 45% of participants ..") discusses only the older evidence and mentions “ambivalent results from the literature”. If the authors are questioning the conclusions of references 16, 17 and 18 they should say so. The recent evidence regarding fetal and maternal outcomes is clear, so the paper's final sentence, “Further research is needed to evaluate maternal and neonatal outcomes after the implementation of the guideline”, seems inappropriate.

We fully acknowledge that the evidence presented in references 16, 17, and 18 clearly demonstrates improved outcomes with induction at 41 weeks and forms the basis for the 2021 Dutch guideline. However, we respectfully disagree with the suggestion that we only discuss older evidence. On the contrary, we focus primarily on recent studies published in 2022, 2023, and 2024. These include large cohort studies that reflect real-world clinical practice and suggest that outcomes associated with induction at 41 weeks may differ from those reported in randomized controlled trials (RCTs). The studies we reflect on are:

29. Bruinsma A, Keulen JK, Kortekaas JC, van Dillen J, Duijnhoven RG, Bossuyt PM, et al. Elective induction of labour and expectant management in late-term pregnancy: A prospective cohort study alongside the INDEX randomised controlled trial. Eur J Obstet Gynecol Reprod Biol X. 2022;16:100165. PubMed PMID: 36262791.

30. Ravelli ACJ, van der Post JAM, de Groot CJM, Abu-Hanna A, Eskes M. Does induction of labor at 41 weeks (early, mid or late) improve birth outcomes in low-risk pregnancy? A nationwide propensity score-matched study. Acta Obstet Gynecol Scand. 2023;102(5):612-25. Epub 20230313. doi: 10.1111/aogs.14536. PubMed PMID: 36915238; PubMed Central PMCID: PMCPMC10072249.

31. Turkmen S, Binfare L. Foeto-Maternal outcomes of pregnancies beyond 41 weeks of gestation after induced or spontaneous labour. Eur J Obstet Gynecol Reprod Biol X. 2024;24:100339. Epub 20240902. doi: 10.1016/j.eurox.2024.100339. PubMed PMID: 39296876; PubMed Central PMCID: PMCPMC11408994.

32. Rydahl E, Declercq E, Juhl M, Maimburg RD. Routine induction in late-term pregnancies: follow-up of a Danish induction of labour paradigm. BMJ Open. 2019;9(12):e032815. PubMed PMID: 31848171.

While RCTs are essential for evaluating clinical interventions under controlled conditions, their applicability to daily practice (particularly after policy implementation) may be limited. This distinction is at the heart of our discussion. Therefore, we believe that further research is warranted to assess maternal and neonatal outcomes following the actual implementation of the guideline, to determine whether the expected benefits of induction are realized in practice.

15. CLINICAL IMPORTANCE: Guidelines are meant to guide, not dictate, clinical practice, but the increase in obstetric litigation in developed countries has increased their clinical importance. It has also introduced an element of defensive medicine exemplified by “Module 2” of the Dutch guideline. I have not read that guideline but the summary in “Textbox 2” of this paper suggests that its recommendations are totally unrealistic. The present paper is important because it explores the mistrust between obstetricians and midwives in the Netherlands and reveals the reluctance of some midwives to accept high-quality evidence. I think the paper would be improved by discussing these issues rather than focussing on the details of the data.

Thank you for your thoughtful and critical reflections. We would like to address three points raised in this comment.

First, with regard to Module 2 of the Dutch guideline and the notion of defensive medicine: while we acknowledge that medico-legal considerations increasingly influence clinical care in many countries, it is not within the scope or intention of our study to assess the validity of the recommendations in the guideline. In our paper, we focused on the perspectives of care providers. We did find that adherence to this specific recommendation was low, which may suggest that it is perceived as somewhat unrealistic.

We now described this in the Discussion section:

“Furthermore, the adherence to recommendations of Module 2 were low, which may suggest that these were perceived as somewhat unrealistic.”

Second, we would like to clarify that our study does not provide evidence of 'mistrust' between midwives and obstetricians. While we did observe differences in perspectives between the two professional groups, we believe these differences reflect the expected variation in professional roles and educational backgrounds. We have discussed these variations in the manuscript, but we would caution against interpreting them as evidence of mistrust.

Third, regarding the suggestion that some midwives show reluctance to accept high-quality evidence, we respectfully disagree. Our findings indicate that Dutch midwives, as part of a highly trained and autonomous profession, deeply value high-quality evidence. However, differences in interpretation may arise, especially when randomized controlled trial outcomes appear to contrast with real-world cohort data or clinical experience. Rather than indicating reluctance, we believe this reflects a thoughtful and reflective weighing of the evidence.

Reviewer 2:

We would like to thank the reviewer for its valuable comments.

16. The scope is narrow for global audience. The objectives which were experiences, perspective and impact on daily work of maternity care providers based on the updated Dutch national guideline is more of an audit of local practice.

Thank you for this comment. We acknowledge that our study, which focuses on the perspectives of maternity care providers regarding the Dutch national guideline, is specific to Dutch practice. However, we believe that the findings are of relevance to a global audience for several reasons.

First, our study identifies factors associated with adherence to the guideline, which may be applicable to the adherence to maternity care guidelines in other countries. As we mention on page 8: “This suggests that awareness, comprehension, and a positive attitude towards guidelines are factors that influence the degree to which guidelines are adhered to.” These factors are not unique to the Netherlands and could be of interest to practitioners and policymakers in other settings.

Second, we observed the influence of profession on agreement, adherence, and opinions regarding the guideline, a finding that is relevant beyond the Dutch context. For example, we noted that maternity care providers working in more medicalized settings, such as hospital-based midwives, tended to hold more medicalized views. This trend is likely to be observed in other countries with similar healthcare settings and structures.

Lastly, our study highlights that adherence to detailed recommendations for shared decision-making can be more challenging than adherence to straightforward recommendations, such as ‘offering induction of labour at 41 weeks of gestation’. This insight is significant for global audiences, as shared decision-making is increasingly emphasized in maternity care worldwide.

While our study is based on the Dutch context, we believe that the insights provided, particularly in the discussion section, offer valuable reflections on the implementation of maternity care guidelines that are relevant to readers from diverse international backgrounds.

17. The outcome of the 2020 systematic review and metanalyses which showed lower incidence of composite adverse perinatal outcome in nulliparous women, has been widely received worldwide and the evidence is superior to the INDEX and SWEPIS studies which the Swedish guideline was based upon. The justification for the study was that the feasibility and acceptability of the guideline was not yet known. This can be better improved upon.

We would like to clarify a few points regarding the evidence and justification for the study.

First, regarding the evidence included in the Dutch guideline, we would like to note that the Dutch guideline was based on a preliminary meta-analysis performed by the working group. The results of the guideline are consistent with the findings of the 2020 systematic review and meta-analysis. Therefore, we believe the guideline was based on evidence that is equally robust. We have added this clarification to the introduction section (page 3):

“The meta-analysis had not yet been published at the time the guideline was developed, but a meta-analysis was performed for the purpose of the guideline.”

Second, the justification for the study was not solely based on the feasibility and acceptability of the guideline. While that was certainly an important aspect, the new policy also raised additional concerns related to hospital capacity, the impact on the daily work of maternity care providers, and adherence to the guideline. These factors were not yet well understood. We have further clarified this in the introduction section (page 3):

“The guideline working group on late-term management noted that the feasibility and acceptability of the recommendations were not yet known. They expressed concern that the new policy may raise issues related to hospital capacity and the ability to provide continuous support during labour [16]. This study aims to explore the experiences and perspectives of maternity care providers on the management of late-term gestations outlined in the guideline. Moreover, it is currently unclear to what extent the guideline is being adhered to. The guideline may also influence daily work of maternity care providers [19], who often face capacity challenges. Understanding adherence and the perspectives and experiences of care providers is crucial to refining the guideline and incorporating valuable feedback from care providers.”

18. The study would have more impact if aspects of the implementation of the guidelines were studied, and the gaps identified in terms of adoption, fidelity, penetration and cost etc. addressed.

We appreciate the reviewer’s suggestion that the study would have more impact if aspects of the implementation of the guidelines were studied, and if gaps in terms of adoption, fidelity, penetration, and cost were addressed. However, we are not entirely clear on the reviewer’s specific expectations regarding these aspects.

In our study, we examined adoption, fidelity, and penetration from the perspective of maternity care providers, which we believed would offer valuable insights into how the guidelines are perceived and implemented in practice. However, we acknowledge that the reviewer may be referring to the need for additional, more objective measures of these factors, such as actual adherence to the guidelines or cost implications. Cost implications of the INDEX-study were studied by Bruinsma et al. (2023). We added a reference to this article in the discussion section:

P9: “Additionally, an analysis of the cost implications of implementing a more restrictive IOL policy is needed, as the cost-effectiveness analysis of the INDEX trial showed that IOL may be cost-effective for nulliparous women but is unlikely to be cost-effective for multiparous women.”

While our study focused on the subjective experiences of care providers, we agree with the reviewer that future research could expand on these findings by incorporating more objective data on the actual impact of the shift in late-term induction policy and its cost implications.

We changed the discussion section as follows:

P7: “In this study, we examined adoption, fidelity, and penetration from the perspective of maternity care providers, which we believed would offer valuable insights into how the guidelines are perceived and implemented in practice. We observed con

---

## [Editor Report · Decision Letter 1]

14 Jul 2025

PONE-D-25-07825R1Maternity care providers’ perspectives on late-term gestation management (LATE-study): a cross-sectional survey.PLOS ONE

Dear Dr. Seijmonsbergen-Schermers,

Thank you for submitting your manuscript to PLOS ONE. After careful consideration, we feel that it has merit but does not fully meet PLOS ONE’s publication criteria as it currently stands. Therefore, we invite you to submit a revised version of the manuscript that addresses the points raised during the review process.

 In the previous decision letter we requested that you upload a copy of the related submission entitled "Shared decision-making on late-term gestation management (LATE-study): a cross-sectional survey among maternity care providers." as an 'other' file when you resubmit your manuscript. We note that in your R2R you state that this has been uploaded but it is not included in the file inventory. Before we can proceed, please upload this as an 'other' file. Please do not remove the current R2R or track changes manuscript in the process. 

We look forward to receiving your revised manuscript.

Kind regards,

Emma Campbell, PhD

Associate Editor

PLOS One

On the behalf of

Patrick Ifeanyi Okonta, MBBCh, MPH, FWACS, FMCOG, MD, DRH

Academic Editor

PLOS ONE
---

## [Author Response · Author response to Decision Letter 2]

15 Jul 2025

Reviewers' comments:

Reviewer 1:

We would like to thank the reviewer for its valuable comments.

10. ORIGINALITY: As pointed out in the Introduction to this paper, there has been a long-standing debate about induction of labour in prolonged pregnancy. Four years ago Drife (Best Practice Res Clin Obstet Gynaecol 2021;77: 3-14) summarised the guidelines at that time as follows:

“A systematic analysis in 2020 identified 49 national and international English-language guidelines which mostly recommended induction between 41and 42weeks [Coates et al, Women Birth 2020;33:219-30]. In the Netherlands, where there is no national guideline, a national survey found substantial differences between midwifery-led and obstetrician-led care [Kortekas et al, BMC Pregnancy and Childbirth 2019;19:181]. A Dutch review in 2018 concluded that evidence was lacking for the recommendation to induce labour at 41 weeks instead of 42 weeks and highlighted the need for more studies [Kuelen et al, Midwifery 2018;66:111-8].”

Such studies were already in progress. The SWEPSIS study (Reference 18 of the present paper) had to be stopped early because of increased perinatal mortality in the expectant management group. That study, and the study of Kuelen et al (Ref 17) along with the systematic review by Alkmark et al (Ref 10), led to a Dutch guideline (Ref 16) developed by midwives, obstetricians, paediatricians and client representatives. The present study is an assessment of how that guideline has been received by the different professional groups in the Netherlands.

Thank you for pointing out the importance of our manuscript.

11. SCIENTIFIC RELIABILITY: I have some general comments but I feel unable to comment on the statistical aspects of the paper as I am not a statistician.

The total of 643 complete or partial responses seems a satisfactory number for analysis but it would be helpful to know what proportion this represents of the total number of midwives and doctors in the areas covered.

Based on an estimated total of 4,120 practicing midwives and 1,200 obstetricians in the Netherlands, the coverage in our study is 11.2% for midwives and 11.6% for obstetricians. We have now included this information in the discussion section of the manuscript (page 10), as follows:

“…(11.6% of all working obstetricians and 11.2% of midwives).”

12. The questionnaire asked participants about agreement with, and adherence to, the guidelines. Adherence was self-reported. Was there any objective measurement of adherence?

As our study was based on a self-administered questionnaire among maternity care providers, adherence to the guideline was self-reported, and no objective measurement was included. We acknowledge this as a limitation. However, we consider the risk of social desirability or recall bias to be limited in this context. Care providers are generally well aware of whether they offer induction at 41 weeks, whether they use decision aids such as choice cards, and adherence to shared-decision making recommendations. Moreover, observational studies may also introduce bias due to altered behavior when being observed. To minimize response bias, the questionnaire was anonymous; we did not collect names or email addresses, allowing participants to respond honestly and confidentially.

13. The “Birth Beliefs Scale” attempts to compare two “beliefs”: one, that childbirth is potentially pathological, and the other that it is physiological. Both are true, so the "beliefs" overlap. What the Scale actually reflects are entrenched attitudes, which are hard to quantify. I'm concerned about the word “beliefs” because it implies that these attitudes will never change.

Thank you for this thoughtful comment. We understand the reviewer’s concern regarding the term “beliefs” and acknowledge that the constructs measured by the Birth Beliefs Scale may also be interpreted as reflecting underlying attitudes toward childbirth. However, the Birth Beliefs Scale is a previously validated instrument, and as such, we have used the original terminology to maintain consistency with validation study. While we agree that alternative wording might better capture the complexity of the concepts, adapting the scale’s terminology is beyond the scope of our study.

14. Attitudes can be changed by evidence, and the 2021 Dutch guideline was written because new evidence showed that the benefits of induction marginally outweigh the risks. In the present paper the paragraph before “Strengths and limitations” (“Furthermore 45% of participants ..") discusses only the older evidence and mentions “ambivalent results from the literature”. If the authors are questioning the conclusions of references 16, 17 and 18 they should say so. The recent evidence regarding fetal and maternal outcomes is clear, so the paper's final sentence, “Further research is needed to evaluate maternal and neonatal outcomes after the implementation of the guideline”, seems inappropriate.

We fully acknowledge that the evidence presented in references 16, 17, and 18 clearly demonstrates improved outcomes with induction at 41 weeks and forms the basis for the 2021 Dutch guideline. However, we respectfully disagree with the suggestion that we only discuss older evidence. On the contrary, we focus primarily on recent studies published in 2022, 2023, and 2024. These include large cohort studies that reflect real-world clinical practice and suggest that outcomes associated with induction at 41 weeks may differ from those reported in randomized controlled trials (RCTs). The studies we reflect on are:

29. Bruinsma A, Keulen JK, Kortekaas JC, van Dillen J, Duijnhoven RG, Bossuyt PM, et al. Elective induction of labour and expectant management in late-term pregnancy: A prospective cohort study alongside the INDEX randomised controlled trial. Eur J Obstet Gynecol Reprod Biol X. 2022;16:100165. PubMed PMID: 36262791.

30. Ravelli ACJ, van der Post JAM, de Groot CJM, Abu-Hanna A, Eskes M. Does induction of labor at 41 weeks (early, mid or late) improve birth outcomes in low-risk pregnancy? A nationwide propensity score-matched study. Acta Obstet Gynecol Scand. 2023;102(5):612-25. Epub 20230313. doi: 10.1111/aogs.14536. PubMed PMID: 36915238; PubMed Central PMCID: PMCPMC10072249.

31. Turkmen S, Binfare L. Foeto-Maternal outcomes of pregnancies beyond 41 weeks of gestation after induced or spontaneous labour. Eur J Obstet Gynecol Reprod Biol X. 2024;24:100339. Epub 20240902. doi: 10.1016/j.eurox.2024.100339. PubMed PMID: 39296876; PubMed Central PMCID: PMCPMC11408994.

32. Rydahl E, Declercq E, Juhl M, Maimburg RD. Routine induction in late-term pregnancies: follow-up of a Danish induction of labour paradigm. BMJ Open. 2019;9(12):e032815. PubMed PMID: 31848171.

While RCTs are essential for evaluating clinical interventions under controlled conditions, their applicability to daily practice (particularly after policy implementation) may be limited. This distinction is at the heart of our discussion. Therefore, we believe that further research is warranted to assess maternal and neonatal outcomes following the actual implementation of the guideline, to determine whether the expected benefits of induction are realized in practice.

15. CLINICAL IMPORTANCE: Guidelines are meant to guide, not dictate, clinical practice, but the increase in obstetric litigation in developed countries has increased their clinical importance. It has also introduced an element of defensive medicine exemplified by “Module 2” of the Dutch guideline. I have not read that guideline but the summary in “Textbox 2” of this paper suggests that its recommendations are totally unrealistic. The present paper is important because it explores the mistrust between obstetricians and midwives in the Netherlands and reveals the reluctance of some midwives to accept high-quality evidence. I think the paper would be improved by discussing these issues rather than focussing on the details of the data.

Thank you for your thoughtful and critical reflections. We would like to address three points raised in this comment.

First, with regard to Module 2 of the Dutch guideline and the notion of defensive medicine: while we acknowledge that medico-legal considerations increasingly influence clinical care in many countries, it is not within the scope or intention of our study to assess the validity of the recommendations in the guideline. In our paper, we focused on the perspectives of care providers. We did find that adherence to this specific recommendation was low, which may suggest that it is perceived as somewhat unrealistic.

We now described this in the Discussion section:

“Furthermore, the adherence to recommendations of Module 2 were low, which may suggest that these were perceived as somewhat unrealistic.”

Second, we would like to clarify that our study does not provide evidence of 'mistrust' between midwives and obstetricians. While we did observe differences in perspectives between the two professional groups, we believe these differences reflect the expected variation in professional roles and educational backgrounds. We have discussed these variations in the manuscript, but we would caution against interpreting them as evidence of mistrust.

Third, regarding the suggestion that some midwives show reluctance to accept high-quality evidence, we respectfully disagree. Our findings indicate that Dutch midwives, as part of a highly trained and autonomous profession, deeply value high-quality evidence. However, differences in interpretation may arise, especially when randomized controlled trial outcomes appear to contrast with real-world cohort data or clinical experience. Rather than indicating reluctance, we believe this reflects a thoughtful and reflective weighing of the evidence.

Reviewer 2:

We would like to thank the reviewer for its valuable comments.

16. The scope is narrow for global audience. The objectives which were experiences, perspective and impact on daily work of maternity care providers based on the updated Dutch national guideline is more of an audit of local practice.

Thank you for this comment. We acknowledge that our study, which focuses on the perspectives of maternity care providers regarding the Dutch national guideline, is specific to Dutch practice. However, we believe that the findings are of relevance to a global audience for several reasons.

First, our study identifies factors associated with adherence to the guideline, which may be applicable to the adherence to maternity care guidelines in other countries. As we mention on page 8: “This suggests that awareness, comprehension, and a positive attitude towards guidelines are factors that influence the degree to which guidelines are adhered to.” These factors are not unique to the Netherlands and could be of interest to practitioners and policymakers in other settings.

Second, we observed the influence of profession on agreement, adherence, and opinions regarding the guideline, a finding that is relevant beyond the Dutch context. For example, we noted that maternity care providers working in more medicalized settings, such as hospital-based midwives, tended to hold more medicalized views. This trend is likely to be observed in other countries with similar healthcare settings and structures.

Lastly, our study highlights that adherence to detailed recommendations for shared decision-making can be more challenging than adherence to straightforward recommendations, such as ‘offering induction of labour at 41 weeks of gestation’. This insight is significant for global audiences, as shared decision-making is increasingly emphasized in maternity care worldwide.

While our study is based on the Dutch context, we believe that the insights provided, particularly in the discussion section, offer valuable reflections on the implementation of maternity care guidelines that are relevant to readers from diverse international backgrounds.

17. The outcome of the 2020 systematic review and metanalyses which showed lower incidence of composite adverse perinatal outcome in nulliparous women, has been widely received worldwide and the evidence is superior to the INDEX and SWEPIS studies which the Swedish guideline was based upon. The justification for the study was that the feasibility and acceptability of the guideline was not yet known. This can be better improved upon.

We would like to clarify a few points regarding the evidence and justification for the study.

First, regarding the evidence included in the Dutch guideline, we would like to note that the Dutch guideline was based on a preliminary meta-analysis performed by the working group. The results of the guideline are consistent with the findings of the 2020 systematic review and meta-analysis. Therefore, we believe the guideline was based on evidence that is equally robust. We have added this clarification to the introduction section (page 3):

“The meta-analysis had not yet been published at the time the guideline was developed, but a meta-analysis was performed for the purpose of the guideline.”

Second, the justification for the study was not solely based on the feasibility and acceptability of the guideline. While that was certainly an important aspect, the new policy also raised additional concerns related to hospital capacity, the impact on the daily work of maternity care providers, and adherence to the guideline. These factors were not yet well understood. We have further clarified this in the introduction section (page 3):

“The guideline working group on late-term management noted that the feasibility and acceptability of the recommendations were not yet known. They expressed concern that the new policy may raise issues related to hospital capacity and the ability to provide continuous support during labour [16]. This study aims to explore the experiences and perspectives of maternity care providers on the management of late-term gestations outlined in the guideline. Moreover, it is currently unclear to what extent the guideline is being adhered to. The guideline may also influence daily work of maternity care providers [19], who often face capacity challenges. Understanding adherence and the perspectives and experiences of care providers is crucial to refining the guideline and incorporating valuable feedback from care providers.”

18. The study would have more impact if aspects of the implementation of the guidelines were studied, and the gaps identified in terms of adoption, fidelity, penetration and cost etc. addressed.

We appreciate the reviewer’s suggestion that the study would have more impact if aspects of the implementation of the guidelines were studied, and if gaps in terms of adoption, fidelity, penetration, and cost were addressed. However, we are not entirely clear on the reviewer’s specific expectations regarding these aspects.

In our study, we examined adoption, fidelity, and penetration from the perspective of maternity care providers, which we believed would offer valuable insights into how the guidelines are perceived and implemented in practice. However, we acknowledge that the reviewer may be referring to the need for additional, more objective measures of these factors, such as actual adherence to the guidelines or cost implications. Cost implications of the INDEX-study were studied by Bruinsma et al. (2023). We added a reference to this article in the discussion section:

P9: “Additionally, an analysis of the cost implications of implementing a more restrictive IOL policy is needed, as the cost-effectiveness analysis of the INDEX trial showed that IOL may be cost-effective for nulliparous women but is unlikely to be cost-effective for multiparous women.”

While our study focused on the subjective experiences of care providers, we agree with the reviewer that future research could expand on these findings by incorporating more objective data on the actual impact of the shift in late-term induction policy and its cost implications.

We changed the discussion section as follows:

P7: “In this study, we examined adoption, fidelity, and penetration from the perspective of maternity care providers, which we believed would offer valuable insights into how the guidelines are perceived and implemented in practice. We observed considerable variat

---

## [Editor Report · Decision Letter 2]

23 Jul 2025

Maternity care providers’ perspectives on late-term gestation management (LATE-study): a cross-sectional survey.

PONE-D-25-07825R2

Dear Dr. Seijmonsbergen-Schermers,

We’re pleased to inform you that your manuscript has been judged scientifically suitable for publication and will be formally accepted for publication once it meets all outstanding technical requirements.

Kind regards,

Patrick Ifeanyi Okonta, MBBCh, MPH, FWACS, FMCOG, MD, DRH

Academic Editor

PLOS ONE
---

## [Editor Report · Acceptance letter]

PONE-D-25-07825R2

PLOS ONE

Dear Dr. Seijmonsbergen-Schermers,

I'm pleased to inform you that your manuscript has been deemed suitable for publication in PLOS ONE. Congratulations! Your manuscript is now being handed over to our production team.

Kind regards,

on behalf of

Professor Patrick Ifeanyi Okonta

Academic Editor

PLOS ONE